# Porphyromonas gingivalis fimbrial protein Mfa5 contains a von Willebrand factor domain and an intramolecular isopeptide

Thomas V. Heidler [1], Karin Ernits[1], Agnieszka Ziolkowska[1], Rolf Claesson[2] & Karina Persson [1✉]

The Gram-negative bacterium *Porphyromonas gingivalis* is a secondary colonizer of the oral biofilm and is involved in the onset and progression of periodontitis. Its fimbriae, of type-V, are important for attachment to other microorganisms in the biofilm and for adhesion to host cells. The fimbriae are assembled from five proteins encoded by the *mfa1* operon, of which Mfa5 is one of the ancillary tip proteins. Here we report the X-ray structure of the N-terminal half of Mfa5, which reveals a von Willebrand factor domain and two IgG-like domains. One of the IgG-like domains is stabilized by an intramolecular isopeptide bond, which is the first such bond observed in a Gram-negative bacterium. These features make Mfa5 structurally more related to streptococcal adhesins than to the other *P. gingivalis* Mfa proteins. The structure reported here indicates that horizontal gene transfer has occurred among the bacteria within the oral biofilm.

[1] Department of Chemistry, Umeå Centre for Microbial Research (UCMR), Umeå University, 90187 Umeå, Sweden. [2] Department of Odontology, Umeå University, 90187 Umeå, Sweden. ✉email: karina.persson@umu.se

Fimbriae (also called pili) are long filamentous protein polymers that project from the bacterial surface and are crucial for attachment to other microorganisms, host cells, and surfaces[1]. They usually contain several protein subunits encoded by the same gene cluster, resulting in the assembly of a long shaft of repetitive proteins decorated by ancillary tip proteins[2–4]. For some gene clusters, flanking transposons have been identified indicating that horizontal gene transfer can occur[5]. In Gram-negative bacteria, these filamentous structures are assembled in a non-covalent manner assisted by multi-protein complexes that span either the outer membrane or both the outer and inner membranes. In contrast, Gram-positive fimbrial proteins are covalently linked to each other by intermolecular isopeptide bonds, which are amide bonds between a lysine side chain of one subunit and the carboxyl group of a C-terminal threonine of the next subunit. The formation of these bonds is mediated by specific sortases that are encoded by the same gene cluster[6,7]. To date six different types of fimbriae have been classified: (1) type-I, also called the chaperone-usher type, (2) type-IV fimbriae, (3) type-IV secretion fimbriae, (4) type-V fimbriae, (5) curli fibers, and (6) sortase-mediated fimbriae, all of which are essential for bacterial virulence[2,4].

*Porphyromonas gingivalis* is a Gram-negative, rod shaped, anaerobic bacteria that belongs to the Bacteroidetes phylum[8]. *P. gingivalis* colonizes the oral biofilm as a secondary colonizer using primary colonizers, mainly commensal streptococci, as attachment partners[9,10]. The proliferation of *P. gingivalis* can lead to a shift from a healthy biofilm composed mainly of commensals to a more pathogenic biofilm[8]. *P. gingivalis* causes chronic inflammation (periodontitis)[11–13] and is believed to be involved in the onset of systemic diseases such as various cancers, rheumatoid arthritis, heart diseases, diabetes, pregnancy complications, and Alzheimer's disease[14–20].

*P. gingivalis* expresses two antigenically distinct but structurally homologous type-V fimbriae that are important both for attachment to other bacteria and for invasion of different host tissues[9,21–25]. FimA fimbriae are up to 2 µm long and Mfa1 fimbriae are ~100-nm long[26,27], and both have been shown to be involved in disease progression[28]. The *mfa1* and *fimA* gene clusters encode five proteins each, including one major stalk subunit (Mfa1 or FimA), a cell-wall anchor (Mfa2 or FimB), and three ancillary tip proteins (Mfa3-5 or FimC-E)[29]. The individual proteins harbor an N-terminal signal peptide that leads them to be transported into the periplasm via the Sec pathway[30]. Within the periplasmic space, the proteins are acylated, processed by signal peptidase II, and transported to the outer membrane by the localization of lipoprotein export pathway[31].

Bacteria of the Bacteroidetes phylum have a specific secretion system, type-IX, which transports proteins and many virulence factors across the outer membrane. Among these are the most important virulence factors in *P. gingivalis*, the gingipains. Those are proteases crucial for processing other virulence factors in addition to causing severe host tissue destruction. Blocking the type-IX secretion system hampers gingipain activity, and is therefore considered an important target for the development of novel antibacterial substances[32]. Proteins translocated by this system are selected by a conserved C-terminal signal domain and among the fimbrial proteins such a C-terminal signal domain is only found in Mfa5. Blocking the type-IX secretion system in *P. gingivalis* inhibits export of Mfa5 to the outer membrane, whereas the other fimbrial proteins are not affected[33], indicating that they are transported over the membrane using another, unknown pathway. Mfa5 is also unique among the fimbrial proteins in that it is exceptionally large (1228 amino acids), and whereas the Mfa1–4 proteins consist of two β-sandwich domains, Mfa5 is predicted to contain several domains, one of which is a von Willebrand factor (vWF) domain.

In this study, we present the crystal structure of the N-terminal half of Mfa5, residues 99–664, at 1.8 Å resolution. Our high-resolution structure reveals unforeseen similarity to adhesins hitherto only observed in Gram-positive bacteria. As predicted, the structure contains a vWF domain with a complete metal ion-dependent adhesion site (MIDAS) as well as two IgG-like domains. Intriguingly, one of the IgG-like domains is stabilized by an intramolecular isopeptide bond, which to our knowledge is the first such bond observed in a Gram-negative bacterial surface protein.

## Results

**Structure determination of Mfa5.** A construct lacking the N- and C-terminal signal peptides, Mfa5$_{21-1044}$, was designed and expressed in *E. coli* (Fig. 1a). The calculated molecular weight was 112 kDa, however, purification resulted in a degradation product estimated to ~70 kDa by SDS-PAGE. A similar fragment was obtained when the protein was treated with α-chymotrypsin. From this sample, intergrown, plate-like crystals were obtained. The crystals diffracted to 1.8 Å, belonged to space group $P2_12_12_1$, and contained one molecule in the asymmetric unit. However, no phase information could be obtained so a shorter construct, Mfa5$_{138-435}$, encompassing only the predicted vWF domain was crystallized. One single rectangular crystal diffracting to 1.85 Å in space group $P2_1$ and with one molecule in the asymmetric unit was obtained. A highly redundant dataset was collected, and the structure was solved using sulfur SAD. The structure was refined to final $R_{work}$ and $R_{free}$ of 13 and 18%. Next, this model was used to solve the structure of the 70 kDa fragment by molecular replacement. A model encompassing amino acids 99–664 could be built into the density and successively refined to $R_{work}$ and $R_{free}$ of 16 and 20%. Final refinement statistics for both structures are presented in Table 1.

**Overall domain architecture.** From the crystallized 70 kDa fragment a model that spans residues 99–664 could be built. The structure consists of three domains (D1 to D3) that build up an elongated shape with dimensions of 46 Å × 48 Å × 164 Å (Fig. 1b). This model will be referred to as Mfa$_{D1-D3}$. The first domain, D1 (amino acids 142–418), forms a Rossman fold classified as a vWF domain (Figs. 2 and 3a)[34]. The vWF domain comprises a cleft harboring a MIDAS motif with a bound Mg$^{2+}$ ion. In addition, a Ca$^{2+}$ ion is coordinated by an adjacent loop region (amino acids 298–303).

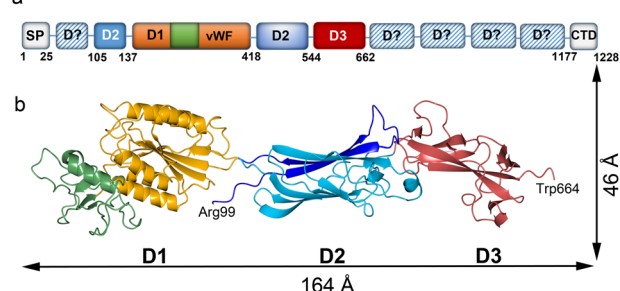

**Fig. 1 Domain arrangement and dimensions of Mfa5. a** A schematic arrangement of the domains of Mfa5 is shown. The D1 domain is depicted in orange (vWF domain) and green (ARM2 extension), the D2 domain is shown in blue, and the D3 domain is shown in red. Potential missing IgG-like domains are shown in striped blue. The N-terminal signal peptide (SP) and the type-IX C-terminal signal domain (CTD) are depicted as gray boxes. **b** Cartoon representation of the Mfa5 crystal structure, residues 99–664. The domains are colored as in (**a**), the domains are numbered and dimensions are indicated.

**Table 1 Data collection and refinement statistics.**

| PDBID | 6tnj | 6to1 |
|---|---|---|
| *Data collection* | | |
| Space group | P2$_1$ | P2$_1$2$_1$2$_1$ |
| *Cell dimensions* | | |
| a, b, c (Å) | 35.20, 110.08, 38.18 | 50.72, 72.81, 184.49 |
| α, β, γ (°) | 90, 98.11, 90 | 90, 90, 90 |
| Wavelength (Å) | 1.5400 | 0.9756 |
| Resolution (Å) | 37.8–1.85 (1.89–1.85) | 47.0–1.79 (1.86–1.79) |
| $R_{merge}$ | 0.087 (0.369) | 0.097 (1.126) |
| $I/\sigma I$ | 75.6 (24.0) | 19.66 (1.87) |
| Completeness (%) | 97.6 (75.9) | 99.45 (95.94) |
| Redundancy | 171.8 (109.8) | 13.1 (11.1) |
| *Refinement* | | |
| Resolution (Å) | 37.8–1.85 (1.90–1.85) | 47.0–1.80 (1.86–1.80) |
| No. reflections | 23902 (1183) | 64104 (2449) |
| $R_{work}/R_{free}$ | 0.133/0.179 | 0.161/0.200 |
| *No. atoms* | | |
| Protein | 2347 | 4461 |
| Ligands/ion | 15 | 17 |
| Water | 389 | 606 |
| *B-factors* | | |
| Protein | 9.92 | 26.76 |
| Ligands/ion | 30.19 | 38.49 |
| Water | 22.29 | 38.57 |
| *R.m.s. deviations* | | |
| Bond lengths (Å) | 0.010 | 0.011 |
| Bond angles (°) | 1.07 | 0.99 |

Statistics for the highest-resolution shell are shown in parentheses. Each structure is based on a single crystal.

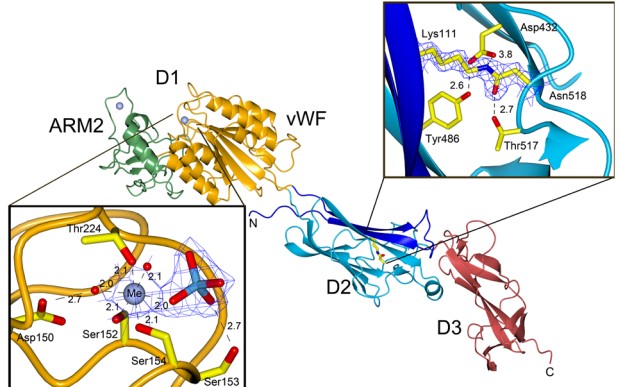

**Fig. 2 Structural features of Mfa5.** Cartoon representation of the Mfa5 model showing the N- and C-termini, the two metal-binding sites (blue spheres), and the isopeptide bond. The domain numbering as well as the vWF domain and the ARM2 domains are labeled. Inserts on the top and bottom show the 2FoFc maps for the isopeptide bond and MIDAS motif at 1.2 sigma, respectively. Distances are shown as dashed lines with values in Ångström, the metal, labeled Me, is shown as a blue sphere, the waters as red spheres and the phosphate ion in red and blue. Side chains are depicted as cylinders.

The second domain, D2 (amino acids 105–135 and 423–544), has an IgG-like fold composed of a β-sandwich with three and five β-strands, respectively. Intriguingly, the vWF domain is inserted between strands B (first sheet) and C (second sheet) of the D2 domain. The overall topology of D2 is similar to CnaA, which is one of the common building blocks of Gram-positive adhesins[35]. Interestingly, an intramolecular isopeptide bond connects Lys111 of the first β-sheet to Asn518 of the second β-sheet (Figs. 2 and 3b).

D3, encompassing amino acids 547–664, comprises six β-strands in a twisted tube-like β-sheet that can be classified as a ubiquitin-like roll. A topology plot shows that there is a similarity to the IgG-like CnaB fold of Gram-positive bacterial adhesins (Figs. 2 and 3c)[35]; however, the generally high B-factor and comparisons to other IgG-like domains led to the assumption that D3 might be incomplete.

**von Willebrand factor domain**. The structure obtained from Mfa5$_{138–435}$, that will be referred to as Mfa5$_{D1}$, is folded into a vWF domain consisting of a typical Rossman fold comprising a six-stranded β-sheet surrounded by three helices on each side. This globular domain, which has a wrench-like form, has the dimensions 44 Å × 52 Å × 79 Å and includes the MIDAS motif, which is a central feature for interaction with various targets. In the Mfa5 MIDAS site, the five residues coordinating the metal ion are situated on the wrench head on top of the β-sheet. Mfa5 has a classical MIDAS motif[36] in which the side chain oxygens of Ser152, Ser154, and Thr224 coordinate the metal directly and the side chains of Asp150 and Asp252 coordinate the metal via a water molecule. The crystallization conditions and the coordination distances (average 2.1 Å) suggest that this is a Mg$^{2+}$ ion (bottom insert in Fig. 2). A homology search with DALI[37] using both Mfa5$_{D1}$ and Mfa5$_{D1-D3}$ indicated a close structural relationship to the Gram-positive tip pilins RrgA (PDBID: 2ww8[38]) and GBS104 (PDBID: 3txa[39]) from *Streptococcus pneumoniae* and *Streptococcus agalactiae* as well as to human integrins such as the collagen-binding α2-I domain (PDBID: 1dzi[40]). A superposition of their respective vWF domains to the vWF domain of Mfa5 gives a good match to the streptococcal tip fimbriae (rmsd 1.9 Å (151 of 296 Cα-positions)) and the α2-I vWF domain (rmsd 2.5 Å (88 of 296 Cα-positions)). This was unexpected based on their low sequence similarity and identity to Mfa5, 19/10% for the streptococcal adhesins and 21/9% for the α2-I protein. A structure-based sequence alignment of the vWF domain confirmed that the MIDAS site is conserved (Supplementary Fig. 1).

Compared to the α2-I structure, the bacterial vWF domain proteins have several additional structural elements. Similar to GBS104 and RrgA[38,39], the Mfa5-vWF domain has a long insertion (ARM2) between the fourth β-strand and the fourth α-helix of the domain (amino acids 257–355). In Mfa5, ARM2 folds into a small domain comprised of long loops centered around a β-sheet of three very short strands. Two helices, separated by a proline, form an L-shaped structure that covers one side of the sheet. One of the loops in the ARM2 domain coordinates the second metal in the structure. The metal is coordinated by the main chain carbonyls of Asn298, Thr301, and Leu303 and by the side chain oxygens of Asn298 and Thr301. In addition, the side chain of Asp336 from another loop also coordinates the metal. The conformation of the loop is additionally constrained by the three prolines Pro299, Pro302, and Pro305. Based on the coordination distances (average 2.4 Å), this metal has been modeled as a calcium ion. Interestingly, the whole ARM2 domain is rich in prolines, 13 of 98 residues, which contributes to the rigidity of the loops (Supplementary Fig. 2). The equivalent ARM2 in RrgA and GBS104 is longer, 123 amino acids (Fig. 4a–d), and does not bind any metals. RrgA and GBS104 also have two additional loops, a 38 amino acid insertion within the β-strand C (ARM1) and a shorter loop on top of α2 (9 amino acids). These two insertions have no equivalents in Mfa5. Nonetheless, in all three bacterial proteins additional loops extend out from the vWF domain and create a lid over the top of the cleft harboring the MIDAS site. Superposing integrin structures bound to extracellular matrix proteins onto Mfa5 shows that the ARM2 domain of Mfa5, in its present

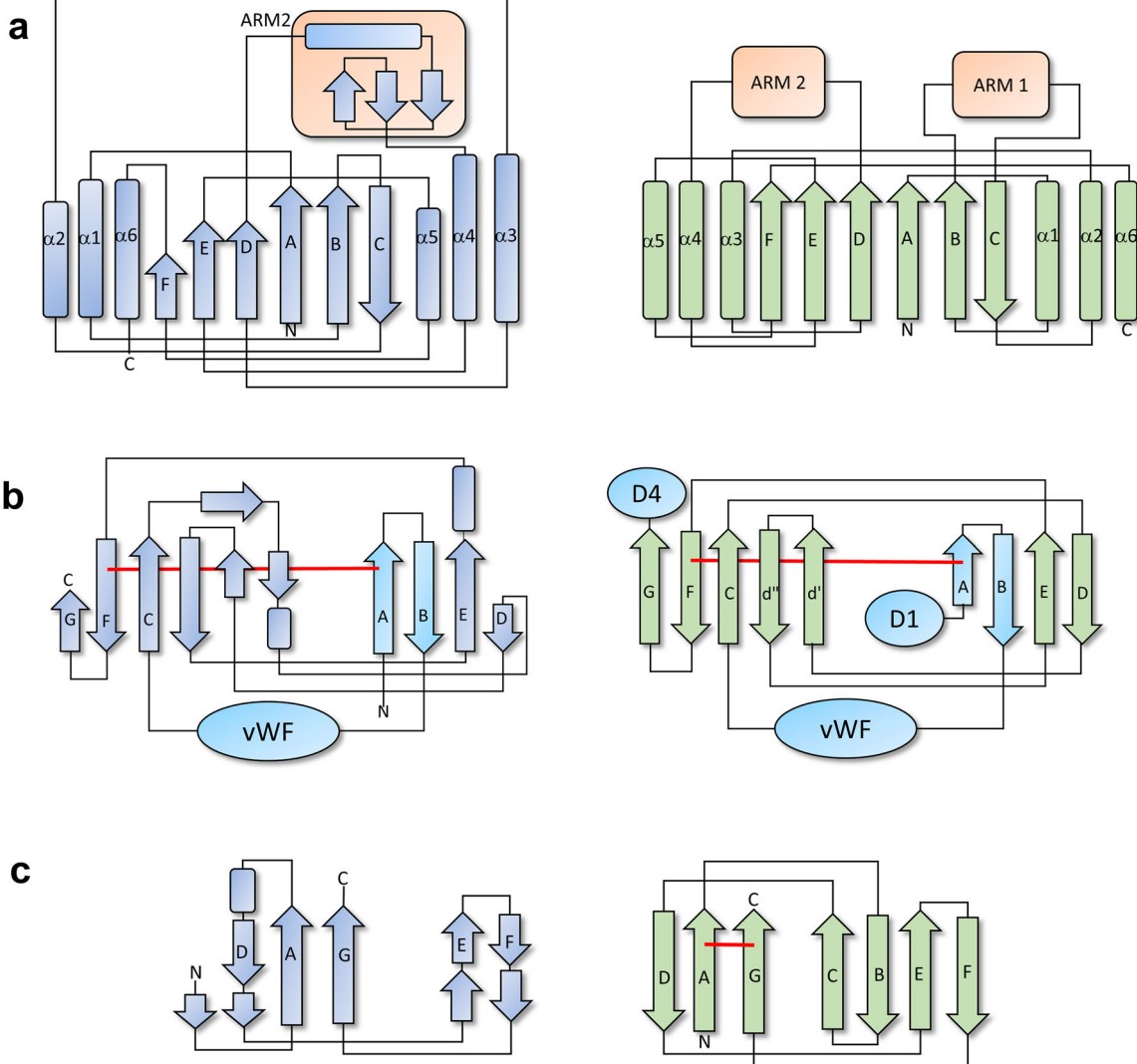

**Fig. 3 Comparison of the Mfa5 topology to Gram-positive fimbrial tip proteins. a** Mfa5 shares the overall fold of the vWF domains and the position of the ARM2 region of the Gram-positive fimbrial adhesins RrgA and GBS104. **b** Mfa5 D2 also shares the CnaA-like fold of GBS104 and RrgA and has the same domain structure. In addition, the positions of the two N-terminal β-strands and the isopeptide bond are identical. **c** In D3 the topology is similar to CnaB-like domains, but β-strands C and B as well as the isopeptide bond are missing. Mfa5 is shown in blue and GBS104 in green. The isopeptide bonds are depicted as red lines. Mfa5 is shown to the left and the streptococcal protein to the right side of the figure. The numbering of the helices and strands are based on what was reported for the GBS104 structure[39].

conformation, would block the collagen binding shown for the integrin α2-I domain (PDBID: 1dzi[40]). In contrast, the ARM2 domain seems to mimic parts of the αV integrin domain, which together with β3 integrin forms a cleft that binds to fibronectin (PDBID: 4mmx[41], Supplementary Fig. 3).

**The isopeptide bond stabilizes the D2 domain**. Although *P. gingivalis* is Gram-negative, the Mfa5 D2 domain has a fold that is common in surface molecules of Gram-positive bacteria. These domains in Gram-positive bacteria are often stabilized by intramolecular isopeptide bonds that are formed spontaneously between the side chains of a lysine and an asparagine or aspartate. Generally, these isopeptide bonds are formed in hydrophobic environments and need an adjacent acid to coordinate the participating side chains of the bond. Surprisingly, in the D2 domain of Mfa5 continuous electron density from the Lys111 ε-amino group to the δ-carboxyamide group of Asn518 is observed, which clearly indicates an intramolecular isopeptide bond (top inlay in

Fig. 2). The surrounding environment is mainly hydrophobic, and the only acid close to the isopeptide bond is Asp432, which is not at hydrogen bonding distance (3.8 and 4.0 Å to the nitrogen and oxygen of the bond, respectively). Instead, the isopeptide bond is stabilized by Thr517 (OG) and by Tyr486 (OH) via a water molecule. An identical arrangement, with an isopeptide bond formed by Lys and Asn residues with coordinating residues Asp, Thr, and Tyr are found in the D2 and N2 domains of RrgA and GBS104, respectively (Supplementary Fig. 2). The intramolecular isopeptide bond in Mfa5 links the two β-sheets of the IgG-like domain; Lys111 is located on the first β-strand of the first β-sheet and Asn518 is located on an antiparallel β-strand of the second β-sheet.

To further validate the presence of the isopeptide bond, purified Mfa5$_{D1-D3}$ and its point mutant K111A were used in ESI-TOF mass spectrometry. This method measures the molecular mass with an accuracy of ±1 Da, and it gave a mass for Mfa5$_{D1-D3}$ of 62,043 Da, which was 15.95 Da less than the theoretical molecular mass of 62,059 Da. This difference in mass indicates the loss of

one $NH_3$ group (17 Da), thus verifying the formation of an isopeptide bond (Supplementary Figs. 4–6). In contrast, the measured mass of 62,002 Da for the Mfa5$_{D1-D3}$ K111A mutant was identical to the theoretical molecular mass. To further analyze if this intramolecular isopeptide bond has an influence on protein stability, a thermal shift assay was performed on both proteins. The Mfa5$_{D1-D3}$ protein showed a two-step unfolding pattern with $T_m$s at 66 and 78 °C. In the K111A mutant, a major peak was observed at 71 °C (Fig. 5). The resulting 7 °C $T_m$ difference and the overall change in unfolding pattern indicate that the absence of the isopeptide bond has a destabilizing effect on the protein.

**Mfa5 integration in the fimbriae.** The *mfa1* gene cluster encodes five proteins, of which Mfa5 has been confirmed as a substrate for the type-IX secretion system[33]. Because native Mfa1 fimbriae from *P. gingivalis* contain Mfa1, 3, 4, and 5 as previously described[42], we decided to derive specific antibodies against Mfa5 and one additional tip protein, Mfa3. When analyzing native fimbriae by SDS-PAGE, the bands at 120–150 kDa and 40 kDa were subsequently verified by western blotting to belong to Mfa5 and Mfa3, respectively (Fig. 6). Using the same Mfa5 antibody serum, in combination with a gold-conjugated secondary antibody and pure native fimbriae for negative staining, a clear Mfa5 tip location was detected. Multiple fimbriae clustered together in one point which was additionally recognized by the secondary antibody (Fig. 7a–c).

## Discussion

Fimbriae are protein polymers projecting from the bacterial surface, and these make the first contact with the targeted host. Therefore, it is of great interest to understand the structure, assembly mechanism, and ligand specificity of fimbriae because their adherence mechanisms and biogenesis are potential targets for the development of novel targeted antibacterials[43]. In recent decades, the combined efforts of many groups have contributed to the understanding of both the ligand specificity and the donor-strand exchange mechanism that underlie the polymerization of type-I fimbriae in Gram-negative bacteria such as *E. coli* (phylum Proteobacteria). This has led to the development of novel antibacterial treatments that have gone as far as clinical trials[4]. *P. gingivalis*, on the other hand, belongs to the Bacteroidetes phylum and accordingly its two fimbriae, Mfa1 and FimA, are of type-V. Recent publications describing the X-ray structures of

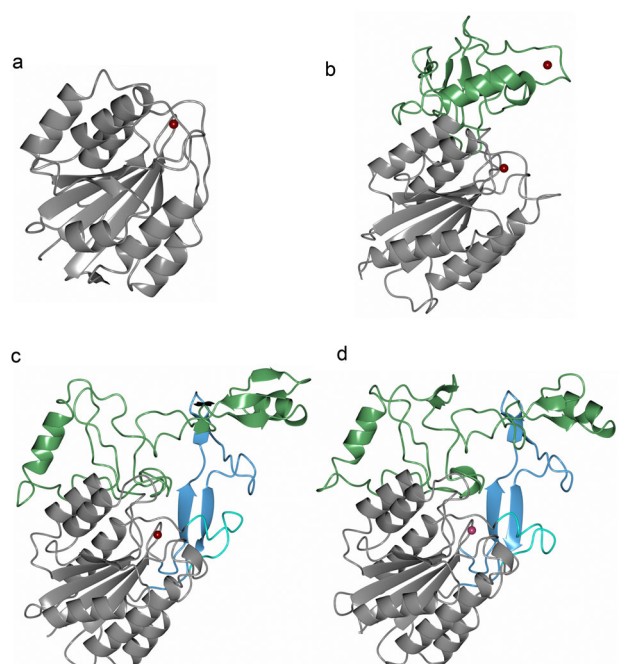

**Fig. 4 Comparison of the ARM subdomains of the vWF domains.** Cartoon representation of the vWF domains of **a** human integrin α2-I domain, **b** *P. gingivalis* Mfa5, **c** streptococcal RrgA, and **d** streptococcal GBS104. The vWF core is colored in gray. The ARM2 regions are colored in green, the ARM1 region in RrgA and GBS104 in blue, and the loop located after α2 in RrgA and GBS104 in cyan. The metal ions are shown as dark red spheres.

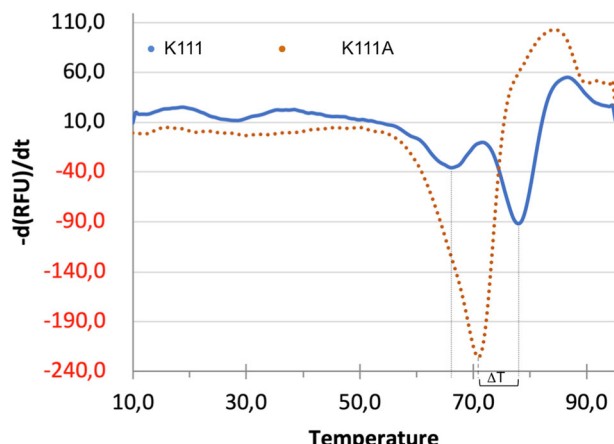

**Fig. 5 The isopeptide bond increases the stability of Mfa5.** The first derivative of the measured fluorescence in a thermal shift assay of Mfa5 (K111) (blue line) and Mfa5(K111A) (orange dotted line). The major unfolding peak is shifted 7 °C when K111 and K111A are compared.

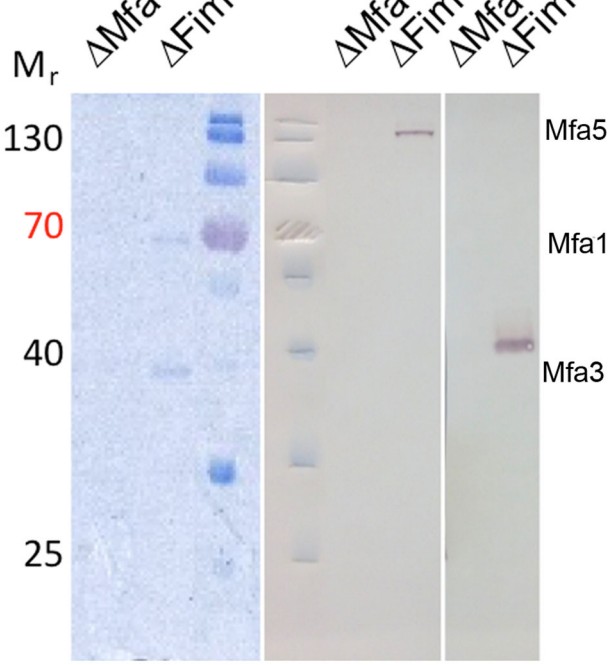

**Fig. 6 Validation of fimbrial proteins.** Antibodies raised against the recombinant Mfa5-vWF domain and Mfa3 were used on native fimbrial purifications. The left panel shows Coomassie-stained bands corresponding to Mfa3 and Mfa1. The middle and right panels show western blots of a fimbrial preparation from Δ*mfa* and Δ*fim* mutants, respectively. The anti-Mfa5-vWF antibody detects a high molecular weight protein, corresponding to Mfa5, in the Δ*fim* mutant, and the anti-Mfa3 antibody detects a protein corresponding to Mfa3 in the Δ*fim* mutant.

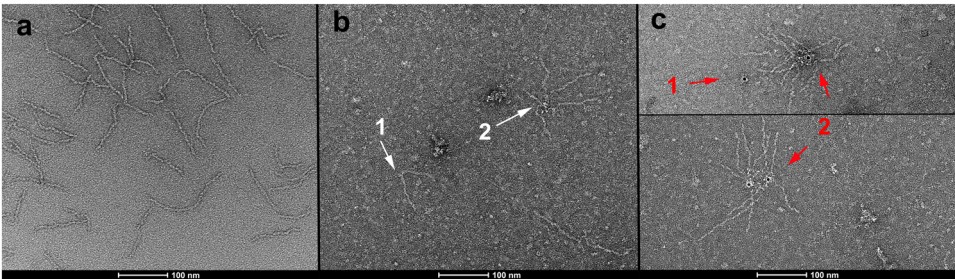

**Fig. 7 Mfa5 localization on the tip.** Negative staining of the native purified Mfa1 fimbriae. **a** Native purified Mfa1 fimbriae. **b** Mfa1 fimbriae in complex with anti-Mfa5-vWF antibodies. The white arrow (1) indicates where an antibody links two fimbriae and arrow (2) indicates where an antibody connects several fimbriae by their tips. **c** Two images of Mfa1 fimbriae in complex with the anti-Mfa5-vWF antibodies and a gold-conjugated secondary antibody. The red arrow (1) indicates a complex between a primary antibody, without attached fimbriae, and the gold-labeled secondary antibody. The red arrows (2) indicate primary antibodies that link several fimbriae. The primary antibodies are complexed with gold-labeled secondary antibodies.

recombinant proteins building up type-V fimbriae describe a common core structure of two domains, both comprising a β-sandwich[44–46] where the first strand of the protein is removed by gingipain proteases when the fimbriae are polymerized. These structures led to the suggestion that type-V fimbriae also depend on a donor-strand exchange mechanism for bioassembly; however, the exact nature of this mechanism is still not fully understood[47]. The *mfa1* gene cluster encodes five proteins where the first four all have the classical type-V fold. The fifth protein, Mfa5, is different from the other Mfa proteins in many ways. It is considerably larger (1228 residues) than Mfa1–4 (324–663 residues), and contrary to Mfa1, 3, and 4 it does not appear to be dependent on gingipains for maturation[33]. Instead, Mfa5 is the only fimbrial protein that has been found to be dependent on the type-IX secretion system for translocation across the outer membrane. Indeed, the crystal structures of the three N-terminal domains of Mfa5, that we present here, do not resemble the other Mfa proteins. The three domains include one vWF domain (D1) and two IgG-like domains (D2 and D3). Surprisingly, despite being unrelated in sequence, Mfa5 shows a striking similarity to the Gram-positive adhesins RrgA and GBS104, which both have a vWF domain and a number of IgG-like domains that appear to function as the stalk to project the VWF domain toward ligands, presumably those presented on host cells. The first Mfa5 IgG-like domain, D2, is formed from two segments of the protein, residues 105–138 that form the first two β-strands and residues 422–546 that constitute the rest of the domain. An intramolecular isopeptide bond between Lys111 and Asn518 links the first segment with the second. Keeping in mind that the Mfa5 D1, D2, and D3 domains presented here only comprise half of the full-length protein, it is possible to imagine that the full-length stalk might be constructed from several additional IgG-like domains. The D3 domain folds like an incomplete IgG-like domain, and comparison to the IgG-like domains in RrgA and GBS104 lead us to speculate that strands are missing from this domain. Hypothetically, extra strands might have been donated by a putative D4 domain if the protein had not been crystallized in a degraded form. In this model, there are also ~75 residues missing from the N-terminus, which may form a segment long enough to contribute with multiple β-strands both to the D2 and D3 domains. Such a segment would hypothetically have the potential to provide more stability to the protein. Furthermore, Mfa5 constitutes the tip of the fimbria together with Mfa3 and Mfa4, and the association of these proteins in the fimbrial spatial arrangement is likely to have a stabilizing effect on the large Mfa5 protein compared to when it is expressed recombinantly on its own. The vWF domain is folded from 276 residues located between the two segments of the D2 domain. vWF domains are most commonly expressed by eukaryotic organisms but have also been found in

some bacteria. In the human integrin α2-I, the vWF domain has its collagen-binding MIDAS motif exposed at the top of the protein, whereas bacterial vWF domains have extra segments (ARM1 and ARM2) folding as subdomains on the sides of the MIDAS motif. The streptococcal adhesins RrgA and GBS104 have two ARMs, whereas Mfa5 only has one. Superposition of the vWF domain of Mfa5 onto the human α2-I domain indicates that ARM2, in the present conformation, would interfere with collagen binding. Superposition of the vWF domain of Mfa5 onto integrin αVβ3 indicates a closer mimic to its fibronectin-binding cleft. Thus the vWF domain is a clear indication that the D1 domain of Mfa5 functions as an adhesin, but the ligand has yet not been determined.

The Mfa5 structure suggests that the ancestor of the *mfa5* gene has been obtained from a streptoccocal ancestor. It has been shown that horizontal gene transfer between microorganisms happens frequently in the oral biofilm where hundreds of species live in close physical contact[48]. Over the course of evolution, Mfa5 has retained crucial structural features from its Gram-positive ancestor, such as the stabilizing intramolecular isopeptide bond and the conserved MIDAS motif, and has adapted to its Gram-negative host by adding a C-terminal domain that targets the protein to the type-IX secretion system, which is unique to the Bacteroidetes phylum. While the assembly mechanism of type-V fimbriae is not fully understood, it is known that the fimbrial proteins, that build up the final fimbria, are expressed as lipidated precursors that are transported to the outer membrane via the lipoprotein export system[31], followed by gingipain-dependent removal of the first β-strand. This cleaved fimbrial protein can then polymerize through a donor-strand replacement mechanism in which a segment from the next fimbrial protein is predicted to participate[44–46]. Integration of Mfa5 into the fimbriae is not expected to follow this mechanism because Mfa5 is not dependent on the presence of gingipains; however, it has been indicated by in vivo studies that the incorporation of Mfa5 into the mature fimbriae requires a functional vWF domain[33]. This needs to be studied further because the truncation of the vWF domain might obstruct the correct formation of the other domains and thus its ability to bind to other fimbrial proteins.

Native fimbrial purifications from *P. gingivalis*, as analyzed by SDS gels, revealed protein bands that were identified as Mfa5 by Hasegawa et al.[42]. In this study, we could further confirm that antibodies derived from recombinant Mfa5 specifically stain these bands. In negative staining images, the antibody against Mfa5 connects multiple fimbriae together, verifying the assumption that Mfa5 is indeed located at the tip of the fimbriae. The different conformations visible in the images also indicate high flexibility, which might be related to the linker regions between the different proteins and their domains. A structure of

the full-length native fimbria would answer the remaining questions about the localization of the different subunits and their biogenesis.

*P. gingivalis* is a key pathogen and is strongly associated with the periodontitis that affects a large part of the population worldwide. The presence of *P. gingivalis* is believed to contribute to the onset of other systemic diseases such as Alzheimer's disease, rheumatoid arthritis, and oral and pancreatic cancer, and its capacity to bind early colonizers of the oral biofilm and to extracellular matrix proteins on host cells is deemed to be an important factor. Therefore, further development of anti-adhesive substances to block the vWF-binding cleft or the development of blocking antibodies will enable future tools to hinder the establishment and proliferation of this pathogen. Hindering the fimbriae from attaching to the primary colonizers of the oral biofilm is crucial because if *P. gingivalis* cannot colonize the mouth it will never cause the chronic inflammation that burdens the immune system and it will not have the opportunity to spread to other, non-oral parts of the body.

## Methods

**Expression and purification of recombinant Mfa5 protein.** The primary Mfa5 sequence (GenBank AUR48966.1) was analyzed for conserved features using the web-based SIGNALP, PROSITE, INTERPRO, and Conserved Domains Database, GOR4, SCRATCH, PSIPRED, and PHYRE2 tools[49–56]. Based on the combined results from all analyses, the full-length Mfa5 was amplified by polymerase chain reaction from the *P. gingivalis* ATCC 33277 genome. Additional constructs, containing amino acids 21–1044, 138–435, 99–664, and 99–664 (K111A) were subsequently cloned by polymerase incomplete primer extension[57] into a pET-ZZ-1a[58] vector with flanking NcoI/XhoI sites (primers are presented in Supplementary Table 1).

The recombinant proteins were expressed in *E. coli* C41 (DE3) (Mfa5$_{21-1044}$) or BL21 (DE3 pLysS) (other constructs). Cells were cultivated to an OD$_{600}$ of 0.8 at 37 °C in 2–4 L Luria Bertani media supplemented with 50 μg·mL$^{-1}$ kanamycin. The temperature was shifted to 20 °C and protein expression was induced by the addition of 0.25 mM IPTG. Cells were harvested 16 h later by centrifugation for 20 min at 4000 × $g$, flash frozen in liquid nitrogen, and stored at −80 °C until further processing.

Cell pellets were resuspended in 50 mL cold phosphate-buffered saline (PBS) containing additional 260 mM NaCl, 10 mM imidazole, 2 mM β-mercaptoethanol, 1% Triton X-100, and protease inhibitors (Pierce) and lysed by sonication. Cell debris was removed by centrifugation at 64,000 × $g$ for 30 min, and the supernatant was applied to Ni-IDA resin (TaKaRa Bio). After two wash steps, the protein was eluted with PBS and 250 mM imidazole. Tobacco Etch Virus protease or α-chymotrypsin at a 1:100 (w/w) ratio was added to the eluted sample, and the mixture was dialyzed against PBS overnight at 4 °C. Next, the sample was applied to a fresh Ni-resin column for re-chromatography. The cleaved protein was concentrated with an Amicon Ultra centrifugal filter (10 kDa cutoff; Millipore) and run on a Superdex200 16/60 column (GE Healthcare) in 20 mM Tris pH 7.4 and 100 mM NaCl. Peak fractions were pooled, concentrated, and stored at −80 °C until further use.

**Cultivation of *P. gingivalis*.** Colonies of the *P. gingivalis* strains JI-1 *fimA*-deletion mutant (Δ*fimA*) and *mfa1*-deletion mutant (Δ*mfa1*), both derived from *P. gingivalis* strain ATCC 33277, growing on blood agar medium were used to inoculate tryptic soy broth (TSB) supplemented with 2.5 g·L$^{-1}$ yeast extract, 5 mg·L$^{-1}$ menadione, 2.5 mg·L$^{-1}$ hemin, and 0.01 mg·L$^{-1}$ DTT overnight as previously described[9]. The cultures were preincubated at 37 °C in an anaerobic chamber with an atmosphere of 85% N$_2$, 5% CO$_2$, and 10% H$_2$. Aliquots of 250 mL of anaerobic TSB were subsequently inoculated with 5 mL (2%) overnight cultures of the two *P. gingivalis* mutants for 16–18 h and further processed.

**Purification of intact fimbriae.** A total of 1 L of *P. gingivalis* culture was harvested by centrifugation at 4000 × $g$ for 20 min. The cells were resuspended in 30 mL 20 mM Tris pH 7.4, 10 mM MgCl$_2$, 1.5 M NaCl, 10% sucrose, 0.1 mM DTT, and DNase and lysed by four passes through a French press at 900 psi. Cell debris was separated by centrifugation at 9000 × $g$ for 10 min, and the supernatant was cleared at 143,000 × $g$ for 90 min. The supernatant was saturated with 50% NH$_4$SO$_4$ at 4 °C, and a follow-up centrifugation at 15,000 × $g$ for 30 min was used to collect the precipitated protein. The pellet was resuspended and dialyzed overnight in 20 mM Tris pH 7.4 and 20 mM NaCl. A 1:2 diluted sample was applied on a HiTrap Q1 column (GE Healthcare) and eluted in a step gradient with 20 mM Tris pH 7.4 and 1 M NaCl. Protein-containing fractions were pooled and applied on a HiPrep Sephacryl S-400 HR 16/60 column (GE Healthcare). Fractions corresponding to the peak were collected and applied to a Superose 6 Increase 10/300 column

(GE Healthcare). Peak fractions were collected, and aliquots were flash frozen in liquid nitrogen for storage at −80 °C.

**Western blotting.** Mfa proteins were detected in fimbrial purifications by western blotting. An SDS gel was loaded with 50 ng detectable protein per lane. After electrophoresis, the gel was incubated together with blotting paper and membrane (0.45 μm nitrocellulose, GE Healthcare) in Towbin buffer (25 mM Tris pH 8.3, 192 mM glycine, 20% (v/v) methanol) for 5–10 min. Next, the blot was assembled in a semi-dry blotter and run at 15 V for 15 min. The membrane was blocked with 5% milk in TBST (20 mM Tris pH 7.6, 150 mM NaCl, 0.05% Tween-20) for 40 min at 37 °C and for 20 min at room temperature. The primary antibodies, anti-Mfa5-vWF or anti-Mfa3 (Agrisera AB), were diluted 1:20,000 in 3% milk in TBST and incubated with the membrane on a shaker for 60 min. After washing for 10 min three times with TBST, the secondary antibody (ab97061, Abcam) was applied at a 1:50,000 dilution in 3% milk in TBST for 60 min. After washing three times with TBST, the bound secondary antibody was detected by chemiluminescence using BCIP/NBT (Promega).

**Crystallization, data collection, and structure determination.** Initial screening was performed using Mfa5$_{21-1044}$ (longest construct) at 12 mg·mL$^{-1}$ and Mfa5$_{138-435}$ (predicted vWF domain) at 18 mg·mL$^{-1}$. Crystallization was performed by sitting-drop vapor-diffusion at 18 °C using commercially available screens from Molecular Dimensions and Hampton Research with a Mosquito robot (TTP Labtech). Mfa5$_{21-1044}$ crystals grew in 12–17% PEG 3350, 80 mM Na$_2$PO$_4$ pH 6, 3 mM CaCl$_2$, and 3 mM MgCl$_2$ and were cryoprotected in mother liquor supplemented with 25% glycerol and flash-cooled in liquid nitrogen. A native dataset was collected at beamline MX14.1 at the Berlin Electron Storage Ring Society for Synchrotron Radiation (Germany). One single crystal of Mfa5$_{138-435}$, grown in 20% PEG 500, 10% PEG 20,000, 0.1 M Sodium HEPES/MOPS pH 7.5, and 0.1 M Glu/Ala/Gly/Lys/Ser-mix was obtained. The crystal was directly flash-cooled without additional cryoprotectants. A high redundancy sulfur single-wavelength anomalous dispersion (SAD) dataset was collected at beamline ID29 at the European Synchrotron Radiation Facility (Grenoble, France)[59]. All datasets were processed with XDS[60], and the sulfur SAD data were combined with BLEND[61]. Phase determination, refinement, and automated model building were performed in the PHENIX suite[62]. The initial solution of Mfa5$_{138-435}$ was used as a molecular replacement model for Mfa5$_{21-1044}$, which was built by BUCCANEER[63] and with cycles of manual building in COOT[64] followed by refinement with PHENIX-refine. The final structures were validated using MOLPROBITY[65] and PDBSUM[66], and analyzed with CATH[34], SALIGN[67], and DALI[37]. Figures were prepared with CCP4MG[68].

**Thermal shift assay.** A stable construct representing the long form of the Mfa5 structure, Mfa5$_{99-664}$ and its isopeptide mutant Mfa5$_{99-664}$ (K111A), were analyzed for thermal stability. The protein samples at 1 mg·mL$^{-1}$ were mixed with Tris buffer pH 8.5 and SYPRO Orange (Molecular Probes), to a final concentration of 20 mM and 5×, respectively. The assay was conducted using a stepwise temperature increase from 10 to 95 °C in a CFX Connect instrument (BioRad) at a rate of 2 °C/min. Data points were collected with the FAM channel (excitation 450–490 nm/detection 510–530 nm) after each 0.5 °C with 3 s equilibration time. Plotting the first derivative of the measured fluorescence at each data point versus the temperature in Microsoft Excel allowed the calculation of the inflection point, and the minima were referred to as the melting temperatures ($T_m$s). Each point was measured in triplicate, and their average value was used.

**Mass spectrometry.** The accurate molecular mass for Mfa5$_{99-664}$ and its K111A mutant was determined by electrospray ionization time of flight (ESI-TOF) mass spectrometry. Purified protein, 4 μL at 10 mg·mL$^{-1}$, was separated on a C8 column using a gradient of 0.1% aqueous formic acid to 0.1% formic acid in acetonitrile over 3 min on an Agilent 1200 HPLC system. The sample was subsequently ionized in an acquisition mode operated at 4 GHz on an Agilent 6230 LC-TOF/MS. The acquired total ion chromatogram was deconvoluted in the MassHunter Qualitative Analysis software (Agilent; V: B.07.00).

**Transmission electron microscopy.** A reaction mixture of purified *P. gingivalis* fimbriae (190 μL with a total of 7.2 μg fimbriae) was incubated 1 h at 4 °C with anti-Mfa5-vWF antibody serum (12 μL) in 20 mM Tris pH 7.4, 100 mM NaCl and applied to Superdex200 10/30 (GE Healthcare). Peak fractions were directly transferred (3.5 μL) to a carbon-coated and glow-discharged copper grid (CF300-Cu, Electron Microscopy Sciences) and incubated for 3 min. The excess sample was removed by blotting, and the grid was washed with two drops of water and stained with 1.5% w/v uranyl acetate (UA) for 30 s. For gold-labeled samples, the sample was transferred to the grid and after 3 min incubation washed in two drops of water. Secondary anti-rabbit 5-nm-gold-conjugated IgG antibody (5 μL) (Abcam) was added directly to the grid and incubated for 5 min. Grid was blotted dry and washed in two drops of water and was further stained with 1.5% UA for 30 s. A Ceta CMOS camera (4k × 4k pixels, FEI) connected to a Talos L120C transmission electron microscope (FEI, Umeå Core Facility for Electron

Microscopy, Sweden) operating at 120 kV was used to examine the negative-stained samples. Images were recorded using the TIA software (FEI).

**Visualization.** Sequence alignments calculated in MULTALIGN[69] were redrawn in ESPRIPT[70] for better visibility.

**Statistics and reproducibility.** The source data underlying Fig. 5 are provided as Supplementary Data 1 in an Excel spreadsheet. The measurements were performed in triplicates.

**Reporting summary.** Further information on research design is available in the Nature Research Reporting Summary linked to this article.

### Data availability

The structure factors and coordinates for the crystal structures reported in this article have been deposited in the Protein Data Bank with accession codes 6TNJ and 6TO1. All relevant data are available from the authors upon request.

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

## Acknowledgements
This work was funded by The Kempefoundations (SMK-1553, SMK-1756.2, and JCK-1918), The Lars Hierta Memorial Foundation, Magn. Bergvall Foundation and the Swedish Research Council (K.P., 2016-05009). Our thanks go to Dr. Linda Sandblad and Dr. Michael Hall (Umeå Core Facility for Electron Microscopy [UCEM], Sweden) and Dr. Irina Iakovleva (Department of Chemistry) for advice on sample preparation and data acquisition and to Prof. Yoshiaki Hasegawa for the *P. gingivalis* strains. We thank Dr. Mikael Lindberg at the Protein expertise platform, Umeå University for the initial cloning and ongoing advice. We thank the beamline scientists at beamlines ID29 (European Synchrotron Radiadion Facility, Grenoble) and MX14.1 (Berliner Elektronenspeicherring-Gesellschaft für Synchrotronstrahlung, Berlin) for their support.

## Author contributions
T.V.H. performed all protein purifications, crystallizations, and crystallography. K.E. and A.Z. performed negative-stained electron microscopy. R.C. performed the cultivation of *P. gingivalis*. K.P. managed the project. T.V.H. and K.P. wrote the manuscript with contributions from R.C. and K.E.

## Funding

## Competing interests
The authors declare no competing interests.
