## [Peer Review File · Communications Biology]

Reviewers' comments:

Reviewer #1 (Remarks to the Author):

This manuscript describes the partial structure of Mfa5, an protein expressed from the Mfa1 type-V fimbriae operon in the Gram-negative bacterium *Porphyromonas gingivalis*. Interestingly, the X-ray structure of Mfa5 reveals a protein architecture that has more in common with Gram-positive fimbriae adhesins than related FimA fimbriae. This includes the presence of an intra-molecular isopeptide bond. These bonds are widely found in Gram-positive, but have never been found in Gram-negative bacteria, making the discovery of one in Mfa5 particularly novel. Mfa5 has high structural homology to Streptococcal adhesins RrgA and GSB104, including a topology whereby amino acid sequences from distant regions of the linear polypeptide come together to form a distinct protein domain. This, along with the presence of an isopeptide bond, leads the authors to speculate that the Mfa5 is the result of a horizontal gene transfer between *Porphyromonas gingivalis* and Gram-positive bacteria within the oral biofilm.

Overall this is a concise, well written manuscript. However, I have some minor comments that I believe would help make the manuscript better for readers:

Pg 4 - Results; chymotrypin typo [chymotrypsin]

Pg 9 – Discussion; The partial D3 domain could also be completed by the missing N-terminal portion of the protein in much the same manner as in RrgA. This possibility should be made clear to the reader.

Pg 11 – Discussion 9 (pg 8 results); The negative staining provides no more than very weak correlative evidence that Mfa5 resides at the tip of the fimbriae. Both images, with and without antibody, show similar arrangements of fimbriae, each with branched and potential head to head arrangements. Also, the differential magnification of the images makes a comparison more problematic. To back up the claim that Mfa5 is located to the tip of the fimbriae I suggest Immunogold labelling be employed to visualise where the anti-Mfa5 antibody is bound to fimbriae. This would provide important evidence about the location of Mfa5 that would greatly enhance the paper. Without this I do not think it is appropriate to use the TEM images to comment on the location of Mfa5.

Pg 24 – Figure 1; It is hard to follow the topology of Mfa5 with the colours used in this figure. The blue and purple are too similar. I suggest another colour with more contrast, as many red green colour blind people will not be able to distinguish between these two colours. This also applies to the orange and yellow selected for domain 1. More contrast is needed. Please include labelling for the N- and C-terminus.

Pg 25 – Figure 2; Please use uniform colours in both Fig 1 and 2. Changing colours adds a layer of complexity to already complex images.

My recommendation is to accept this manuscript, but with minor changes outlined above. I do believe providing evidence of the location of Mfa5 to the tip of the fimbriae (or another location) is important for the manuscript to provide a biological element to complement the structural component.

Reviewer #2 (Remarks to the Author):

In this clear, concise and beautifully-written communication the authors present an extremely interesting finding that is highly relevant to our understanding of bacterial adhesion in oral biofilms. It thus has important implications for human health.

The subject of the article is the bacterium *Porphyromonas gingivalis*, which is a component of oral biofilms along with commensal *Streptococci*, and whose incorporation can generate pathogenicity, leading to cancers and various other systemic diseases. *P. gingivalis* expresses two types of fimbriae, used in adhesion. These are encoded by two gene clusters whose protein products are used to assemble the fimbriae. One of the proteins, Mfa5, is of particular interest because it is uniquely different from the others; it is substantially larger and is exported by a different pathway. This is the subject of this article.

Heidler et al. have determined the high-resolution crystal structure of a 3-domain N-terminal fragment of Mfa5, comprising the adhesive von Willebrand factor (vWF) domain together with two Ig-like domains. The striking finding is that this protein fragment is structurally homologous with fimbrial adhesins from several Gram-positive streptococci, despite only slight sequence similarity. The vWF domain has a MIDAS site essentially identical to those in RrgA (*S. pneumoniae*) and GBS104 (*S. agalactiae*) and a large insertion (ARM2) at the same position to those in RrgA and GBS104. Most intriguingly, the first Ig-like domain, D2, has an internal Lys-Asn isopeptide bond crosslink at exactly the same site as in RrgA and GBS104. Such internal isopeptide bonds are common in the cell-surface fimbriae of Gram-positive organisms but have never been seen before for any Gram-negative organism. The inescapable conclusion is that Mfa5 has been acquired by horizontal gene transfer from other components of the oral biofilm (presumably *Streptococci*).

This is a highly original finding that will be of wide interest across the fields of microbial pathogenesis, biochemistry and structural biology. The article makes very interesting reading, is nicely illustrated, and all the experimental work (protein expression, crystallography, mass spectrometry, thermal shift analysis etc) is well documented so that it could be replicated by others. The conclusions are clear, and it opens the way towards new therapeutic approaches targeting pathogenic biofilms.

The work is complete in itself, and does not require any further experimentation. I have only a few very minor comments:

1. In the abstract, last line: Missing word. "...bacteria that reside within the...."
2. In the Introduction, pp 3-4, Does blocking the type-IX secretion system (and hence Mfa5 export) have any impact on virulence?

3. On p.4 line 17. (spelling) chymotrypsin

4. I agree that, from the data given here, the metal ion in the MIDAS site is almost certainly Mg²⁺. On p.6 it is stated that the second metal ion, bound in an external loop, is probably Ca²⁺, based on the metal-ligand distances. Could these be given in the text please?

Reviewer #3 (Remarks to the Author):

Heidler et al. report the high-resolution X-ray structure of the N-terminal half of the Mfa5 tip adhesin subunit of type-V fimbriae. This fragment contains the most interesting functional part of the subunit, a von Willebrand factor domain and two adjacent IgG-like domains. Surprisingly, the structure appeared to have similarity to adhesins from Gram-positive bacteria. Moreover, one of the IgG-like domains is stabilized by an intramolecular isopeptide bond, which has been observed in adhesins from Gram-positive bacteria, but not in Gram-negative bacterial surface proteins. In my opinion, this is an excellent structural study. The structure is of high quality judging by the statistics provided. It is well described both in the text and figures and the authors made a good structural bioinformatics study. Hence, I do not have any specific comments. Unfortunately, although the paper reports the structure of the Mfa5 fragment that is to be important for the fimbria adhesion, it says very little about the mechanism of adhesion itself. Hence, this study is somewhat of low value from the biological point of view.

Author Reply

We are very grateful for the comments and suggestions we received on our manuscript "**Mfa5 from *Porphyromonas gingivalis*: a von Willebrand factor domain and an intramolecular isopeptide bond in a Gram-negative bacterial fimbrial protein**" and feel that in addressing these issues our manuscript has been significantly improved. Most importantly we have used immunogold labeling to visualize where the anti-Mfa5 antibody is bound to the Mfa fimbriae. In order to perform the immunogold labelling two contributing authors have been added, Karin Ernits and Agnieszka Ziolkowska.

In addition to the comments from the reviewers we have also updated Fig 3 (Topology diagrams) since we saw that some of the labeling was done with different fonts.

Our replies to the reviewers' comments:

Reviewer #1 (Remarks to the Author):

This manuscript describes the partial structure of Mfa5, a protein expressed from the Mfa1 type-V fimbriae operon in the Gram-negative bacterium *Porphyromonas gingivalis*. Interestingly, the X-ray structure of Mfa5 reveals a protein architecture that has more in common with Gram-positive fimbriae adhesins than related FimA fimbriae. This includes the presence of an intra-molecular isopeptide bond. These bonds are widely found in Gram-positive, but have never been found in Gram-negative bacteria, making the discovery of one in Mfa5 particularly novel. Mfa5 has high structural homology to Streptococcal adhesins RrgA and GSB104, including a topology whereby amino acid sequences from distant regions of the linear polypeptide come together to form a distinct protein domain. This, along with the presence of an isopeptide bond, leads the authors to speculate that the Mfa5 is the result of a horizontal gene transfer between *Porphyromonas gingivalis* and Gram-positive bacteria within the oral biofilm.

Overall this is a concise, well written manuscript. However, I have some minor comments that I believe would help make the manuscript better for readers:

Pg 4 - Results; chymotrypin typo [chymotrypsin]

Reply: This has been corrected.

Pg 9 – Discussion; The partial D3 domain could also be completed by the missing N-terminal portion of the protein in much the same manner as in RrgA. This possibility should be made clear to the reader.

Reply: The possibility that the N-terminal portion may run through both the D2 and D3 domains is now discussed (discussion section).

Pg 11 – Discussion 9 (pg 8 results); The negative staining provides no more than very weak correlative evidence that Mfa5 resides at the tip of the fimbriae. Both images, with and without antibody, show similar arrangements of fimbriae, each with branched and potential head to head arrangements. Also, the differential magnification of the images makes a comparison more problematic. To back up the claim that Mfa5 is located to the tip of the fimbriae I suggest Immunogold labelling be employed to visualise where the anti-Mfa5 antibody is bound to fimbriae. This would provide important evidence

about the location of Mfa5 that would greatly enhance the paper. Without this I do not think it is appropriate to us the TEM images to comment on the location of Mfa5.

Reply: We have now prepared more negative stain experiments on Mfa fimbriae treated with anti Mfa5 antibody with and without secondary antibody. We can clearly see that several Mfa fimbria are attached to the antibody (IgM) by their end parts and that the connecting antibody is labelled by the gold-labelled antibody. We would like to emphasize to the reviewers that the experiment is performed with serum and that we have a mix of both IgG and IgM, which explains the many fimbriae connected.

Pg 24 – Figure 1; It is hard to follow the topology of Mfa5 with the colours used in this figure. The blue and purple are too similar. I suggest another colour with more contrast, as many red green colour blind people will not be able to distinguish between these two colours. This also applies to the orange and yellow selected for domain 1. More contrast is needed. Please include labelling for the N- and C-terminus.

Reply: Figure 1 has been changed, N- and C-terminus are labeled with Arg99 and Trp664 respectively, and the colors are now more distinct.

Pg 25 – Figure 2; Please use uniform colours in both Fig 1 and 2. Changing colours adds a layer of complexity to already complex images.

Reply: Figure 2 has now been changed according to this suggestion.

My recommendation is to accept this manuscript, but with minor changes outlined above. I do believe providing evidence of the location of Mfa5 to the tip of the fimbriae (or another location) is important for the manuscript to provide a biological element to complement the structural component.

Reviewer #2 (Remarks to the Author):

In this clear, concise and beautifully-written communication the authors present an extremely interesting finding that is highly relevant to our understanding of bacterial adhesion in oral biofilms. It thus has important implications for human health.

The subject of the article is the bacterium *Porphyromonas gingivalis*, which is a component of oral biofilms along with commensal *Streptococci*, and whose incorporation can generate pathogenicity, leading to cancers and various other systemic diseases. *P. gingivalis* expresses two types of fimbriae, used in adhesion. These are encoded by two gene clusters whose protein products are used to assemble the fimbriae. One of the proteins, Mfa5, is of particular interest because it is uniquely different from the others; it is substantially larger and is exported by a different pathway. This is the subject of this article.

Heidler et al. have determined the high-resolution crystal structure of a 3-domain N-terminal fragment of Mfa5, comprising the adhesive von Willebrand factor (vWF) domain together with two Ig-like domains. The striking finding is that this protein fragment is structurally homologous with fimbrial adhesins from several Gram-positive streptococci, despite only slight sequence similarity. The vWF domain has a MIDAS site essentially identical to those in RrgA (*S. pneumoniae*) and GBS104 (*S. agalactiae*) and a large insertion (ARM2) at the same position to those in RrgA and GBS104. Most intriguingly, the first Ig-like domain, D2, has an internal Lys-Asn isopeptide bond crosslink at exactly the same site as in RrgA and GBS104. Such internal isopeptide bonds are common in the cell-surface fimbriae of Gram-positive organisms but have never been seen before for any Gram-negative organism. The inescapable conclusion is that Mfa5 has been acquired by horizontal gene transfer from other components of the oral biofilm (presumably *Streptococci*).

This is a highly original finding that will be of wide interest across the fields of microbial pathogenesis, biochemistry and structural biology. The article makes very interesting reading, is nicely illustrated, and all the experimental work (protein expression, crystallography, mass spectrometry, thermal shift analysis etc) is well documented so that it could be replicated by others. The conclusions are clear, and it opens the way towards new therapeutic approaches targeting pathogenic biofilms.

The work is complete in itself, and does not require any further experimentation. I have only a few very minor comments:

1. In the abstract, last line: Missing word. "...bacteria that reside within the...."

Reply: This has been corrected.

2. In the Introduction, pp 3-4, Does blocking the type-IX secretion system (and hence Mfa5 export) have any impact on virulence?

Reply: This is indeed an interesting comment. Blocking the type-IX secretion system blocks the secretion of several virulence factors. The most important virulence factors in *P. gingivalis* are the arginine- and lysine gingipains. These are important for degrading a range of host tissues and proteins as well as the maturation of fimbriae. Since gingipains are secreted via the type-IX secretion system, blocking of this secretion system blocks both the transport of Mfa5, the maturation of the other fimbrial proteins as well as all the other functions that gingipains have. Albeit very interesting, we have added this to the introduction but we have tried to keep it short. Here we have also added a reference (Lasica, A.M., Ksiazek, M., Madej, M. & Potempa, J. *The Type IX Secretion System (T9SS): Highlights and Recent Insights into Its Structure and Function. Front Cell Infect Microbiol* **7**, 215 (2017)

3. On p.4 line 17. (spelling) chymotrypsin

Reply: Corrected.

4. I agree that, from the data given here, the metal ion in the MIDAS site is almost certainly Mg²⁺. On p.6 it is stated that the second metal ion, bound in an external loop, is probably Ca²⁺, based on the metal-ligand distances.

Reply: The average distances are now given in the text.

Reviewer #3 (Remarks to the Author):

Heidler et al. report the high-resolution X-ray structure of the N-terminal half of the Mfa5 tip adhesin subunit of type-V fimbriae. This fragment contains the most interesting functional part of the subunit, a von Willebrand factor domain and two adjacent IgG-like domains. Surprisingly, the structure appeared to have similarity to adhesins from Gram-positive bacteria. Moreover, one of the IgG-like domains is stabilized by an intramolecular isopeptide bond, which has been observed in adhesins from Gram-positive bacteria, but not in Gram-negative bacterial surface proteins. In my opinion, this is an excellent structural study. The structure is of high quality judging by the statistics provided. It is well described both in the text and figures and the authors made a good structural bioinformatics study. Hence, I do not have any specific comments. Unfortunately, although the paper reports the structure of the Mfa5 fragment that is to be important for the fimbria adhesion, it says very little about the mechanism of adhesion itself. Hence, this study is somewhat of low value from the biological point of view.

REVIEWERS' COMMENTS:

Reviewer #1 (Remarks to the Author):

I am happy to recommend that this revised article be accepted for publication.